# Impact of the Digital Economy on PM_2.5_: Experience from the Middle and Lower Reaches of the Yellow River Basin

**DOI:** 10.3390/ijerph192417094

**Published:** 2022-12-19

**Authors:** Huaxue Zhao, Yu Cheng, Ruijing Zheng

**Affiliations:** College of Geography and Environment, Shandong Normal University, Jinan 250358, China

**Keywords:** digitization, haze pollution, mediating effect model, spatial Durbin model, kernel density estimation

## Abstract

The development of the digital economy holds great significance for alleviating haze pollution. To estimate the impact of the digital economy on haze pollution, this paper explores the spatiotemporal evolutionary characteristics of the digital economy and PM_2.5_ concentration in the middle and lower reaches of the Yellow River Basin from 2011 to 2019 and conducts regression analysis by combining a fixed effect (FE) model and the spatial Durbin model (SDM). Moreover, this study divides the mitigation effect of haze pollution into a direct effect and a spatial spillover effect, and it further analyzes the mechanism from the perspectives of technological innovation and the industrial structure. The empirical results show that the development level of the digital economy increases year by year and that the concentration of PM_2.5_ decreases year by year. The digital economy level and PM_2.5_ concentration in the downstream region are higher than those in the middle region, and the digital economy is negatively correlated with haze pollution. Similarly, the spatial spillover effect of the digital economy is conducive to curbing haze pollution. The robustness test also supports this conclusion. In addition, there is regional heterogeneity in the impact of the digital economy on haze pollution. The direct effect and spatial spillover effect of the digital economy on haze pollution in the downstream region are greater than those in the middle region. This study suggests that to realize air pollution prevention and control, it is necessary to strengthen the construction of digital infrastructure and create a good digital economy development environment based on local conditions. Encouraging the development of digital technological innovation and promoting industrial digital transformation hold great significance for alleviating haze pollution.

## 1. Introduction

The economic development model of the Yellow River basin has long been dominated by extensive heavy industry. Traditional industries such as the energy, chemical engineering, raw material processing and animal husbandry industries in the middle and lower reaches of the region account for a large share, and innovation, greening, and strong competitive emerging industries are lacking. The development of traditional industries has caused some obstacles to improving air quality in the Yellow River Basin. In 2020, the PM_2.5_ concentrations of the middle and lower reaches of the basin were approximately 15% higher than the national average. In addition, the ‘2020 national eco-environmental quality profile’ indicated that 20 cities had poor air pollution, of which 15 were located in the region. Therefore, air quality issues have become a focus [1]. To improve air quality, strengthen ecological and environmental protection, and focus on improving the ecological and environmental quality of the middle and lower reaches of the Yellow River Basin, the ‘Yellow River Basin Ecological and Environmental Protection Plan’ proposes to eliminate weather with heavy pollution by 2025 and achieve air quality standards in the whole river basin by 2030 [2]. Thus, there is an urgent need to solve the problem of haze pollution.

At present, the digital economy is developing in depth, its scale is expanding, and its contribution is increasing. In 2019, the value added to China’s digital economy reached 35.8 trillion yuan, accounting for 36.2% of GDP [3]. In the same year, the scale of digital economy development in the provinces along the Yellow River Basin exceeded 7 trillion yuan, accounting for more than 20% of the national total. The level of digital economy development in Shandong and Henan Provinces was much higher than that in other provinces [4]. With the development of digital technologies such as big data, cloud computing and artificial intelligence, the digital economy has played a role in improving the quality and efficiency of environmental protection, economic development and pollution control by optimizing the industrial structure and improving innovation efficiency in the Yellow River Basin [5].

In this context, exploring the impact of the digital economy on haze pollution is crucial to promoting haze control and sustainable development in the Yellow River Basin. Therefore, the following questions remain to be clearly answered: How does the digital economy affect haze pollution? What is the specific transmission path of the effect of the digital economy on haze pollution? In addition, the spatial spillover effect of the digital economy on haze pollution and the spatial heterogeneity of the effect deserve attention. Based on 2011–2019 panel data on 57 cities in the middle and lower reaches of the Yellow River Basin, this study constructs a panel regression model to verify the inhibitory effect of the digital economy on haze pollution. In addition, this paper focuses on the multidimensional path of the effect of the digital economy on haze pollution, which involves technological innovation and the industrial structure. Finally, this paper also uses the spatial Durbin model (SDM) to consider the spatial spillover effect of the digital economy on haze pollution and the spatial heterogeneity of the effect. To ensure the reliability of the results, we replaced the spatial weight matrix and the core explanatory variable lagged by one period to conduct the robustness test. Effectively evaluating the control effect of the digital economy on air pollution can provide policy support for air pollution prevention and control and further promote the ecological environment and high-quality development of the Yellow River Basin.

The remainder of this paper is organized as follows. Section 2 is a literature review. Section 3 proposes a theoretical model to explore the relationship between the digital economy and haze pollution, and it describes the relevant data and their sources. Section 4 presents the empirical results and discussion. Section 5 summarizes the full text and proposes countermeasures.

## 2. Literature Review

The problem of haze pollution reflects the condition of air quality, which affects people’s health and is closely related to people’s productivity and life [6,7]. At present, studies of haze pollution mainly focus on its spatial evolutionary characteristics, spatial differences [8], health effects [9,10] and influencing factors [11,12]. Many scholars in China and elsewhere have found that technological innovation [13], the industrial structure [14], urbanization [15], environmental regulation [16], openness [17] and financial development [18] are significant factors affecting haze pollution, while research on the influence of the digital economy on haze pollution mainly considers the path and spatial effect.

The digital economy, as the main economy following the agricultural economy and the industrial economy, is divided into two categories: The first is the industry directly related to information and communication, namely, digital industrialization, and the second is the integration of traditional industries and digital technologies, namely, industrial digitalization. Studies in China and elsewhere mainly focus on the economic effects of information and communication technology (ICT). On the one hand, the digital economy promotes economic growth from the perspective of industrial structure, and research finds that the digital economy will bring industrial innovation effects, industrial integration effects, and industrial correlation effects and ultimately promote industrial restructuring, transformation, and upgrading [19,20]. Guan et al. [21] pointed out that the digital economy can accelerate the transformation and upgrading of the industrial structure by stimulating regional innovation. On the other hand, the digital economy can promote macroeconomic growth by optimizing the allocation of factors and raising the level of total factor productivity (TFP). Some scholars believe that the development of the digital economy can significantly improve the allocation of data elements and achieve the purpose of improving traditional production efficiency and promoting economic growth through the integration of data elements with single production factors such as labor and capital [22,23]. The measurement of TFP includes all production factors such as labor, capital, energy, and raw materials in the production process. It is an indicator to measure the efficiency of economic growth. Industrial structure upgrading and technological innovation are two key paths for the digital economy to improve total factor productivity. For example, Liu et al. [24] found that the digital economy significantly improved China’s green TFP (GTFP) through industrial structure upgrading by using Tobit, quantile regression, and mediating effect models. Pan et al. [25] found that the digital economy has an innovation-driven effect on China’s TFP and promotes the extensive and sustainable development of TFP. Although previous studies have shown that the digital economy has an impact on environmental pollution, no consistent conclusion has been reached due to the differences in research methods, research objects and research perspectives. The first view is that the digital economy has exacerbated environmental pollution. The consumption of production resources, energy consumption and the disposal of digital product waste will cause enormous damage to the ecological environment, especially in developing countries [26]. Yan et al. [27] investigated the impact of the development of ICT on energy productivity, and the results showed that the development of ICT consumed much energy and caused environmental pollution. Anders et al. [28] estimated the global electricity use of communication equipment from 2010 to 2030 and concluded that ICT energy consumption will contribute approximately 23% of global greenhouse gas (GHG) emissions in 2030. Perry Sadorsky [29] and Steffen Lange et al. [30] also argued that the use of digital products will increase electricity consumption and pollutant gas emissions. Salahuddin and Alam [31] came to a similar conclusion. To sum up, the main reasons why the digital economy aggravates environmental pollution are the energy consumption in the production process of many digital products and the pollution caused by the degradation and treatment of digital waste.

Another view holds that the digital economy has a nonlinear impact on environmental pollution and can improve the ecological environment and reduce pollutant emissions through industrial structure optimization and technological innovation [32,33]. Yu [34] considered the influence of the Internet on electricity intensity based on dynamic panel regression, spatial Durbin and other models. The results showed that the development of the Internet has a significant negative spatial spillover effect on power intensity. Lin [35] and Li [36] took Chinese provincial data as samples and proved that the development of the Internet could achieve energy conservation and emission reduction through industrial structure upgrading and technology diffusion; additionally, this impact was nonlinear. This conclusion still holds for haze pollution. The digital economy has a negative impact on haze pollution, but the restraining effect is nonlinear and enhanced with the improvement of the development level of the digital economy. Technological innovation, industrial structure upgrading, and green development are the three main mechanisms for digital finance to reduce smog pollution [37]. Yang et al. [38] explored the nonlinear mechanism of digital finance and haze pollution and found that a high level of economic development would help alleviate air pollution. Due to the mobility of PM_2.5_ and the technology and knowledge spillovers of digital economy development, the digital economy has a strong spatial effect on haze pollution. Zhou [39] and Deng [40] proved that the digital economy has a spatial inhibitory effect on haze pollution from the perspective of space. Qi et al. [41] also reached a similar conclusion. In addition, some scholars argue that with the development of the Internet and other digital technologies, the increasing attention paid to environmental issues by social media has significantly alleviated smog pollution [42,43].

In summary, studies show that the problem of haze pollution has aroused widespread concern. However, most studies discuss the impact of digital finance, the Internet and ICT on environmental pollution, and few studies focus on the inhibitory effect of the digital economy on haze pollution. In addition, existing research lacks the influence mechanism and spatial effect analysis. At the same time, most research takes the country as the study area, and there is a lack of research at the small scale of typical regions. The innovation of this paper is that it constructs a theoretical model to describe the impact mechanism and spatial effect of the digital economy on haze pollution based on the middle and lower reaches of the Yellow River Basin, where haze pollution is relatively serious. This paper discusses the mechanism of the effect of the digital economy on haze pollution based on two aspects, technological innovation and the industrial structure. Finally, the spatial heterogeneity of the effect of the digital economy on haze pollution is discussed.

## 3. Methodology and Data

### 3.1. Study Area

The Yellow River Basin flows through nine provinces, including Qinghai, Sichuan, Gansu, Ningxia, Inner Mongolia, Shanxi, Shaanxi, Henan and Shandong. It is an important ecological security barrier in China and an important region for population activities and economic development. Especially in the middle and lower reaches of the basin, traditional industries with high energy consumption, high emissions and high pollution are clustered, and the air pollution problem urgently needs to be solved. Based on the division standard of the Yellow River Water Conservancy Commission in this study basin area, a total of 57 cities are included as the research object (because of the lack of data on Jiyuan city and Laiwu city in Jinan in 2019, the two cities are not included within the scope of this research) (Figure 1).

### 3.2. Model Setting

#### 3.2.1. Kernel Density Estimation

A kernel density estimation (KDE) model can reveal the probability density of random variables and uses continuous density curves to describe the distribution form of random variables. The location, shape and ductility of the variable distribution can be reflected by a KDE model [44]. The formula is as follows:(1)$$\;\;f\left(x\right)=\frac{1}{Nh}\overset{n}{\underset{i=1}{\sum }}K\left(\frac{x_{i}-\bar{x}}{h}\right)$$
where, $x_{i}$ is the observed value of x variable in city i, $\bar{x}$ is the mean value, N is the number of observations, h is the bandwidth, and K(*) is the kernel function. In this study, we selectde the most frequently used Gaussian kernel function to study the dynamic evolution of the digital economy and PM_2.5_ concentration. The formula of Gaussian kernel function is:(2)$$K\left(x\right)=\frac{1}{\sqrt{2\pi }}exp\left(-\frac{x^{2}}{2}\right)$$

#### 3.2.2. Benchmark Regression Model

According to previous research and theoretical analysis, the digital economy has a certain impact on haze pollution. Therefore, we used a fixed effect (FE) model to verify the direct impact of the digital economy on haze pollution in the middle and lower reaches of the Yellow River Basin. The formula is as follows:(3)$$lnPM_{2.5_{i,t}}=\alpha_{0}+\alpha_{1}lnDE_{i,t}+\alpha_{2}Controls_{i,t}+u_{i}+v_{i}+\mu_{it}$$
where $lnPM_{2.5_{i,t}}$ and $lnDE_{i,t}$ are indicators of the PM_2.5_ concentration and the development level of the digital economy in city i in period t, respectively; $Controls_{i,t}$ is a series of related control variables; α represents the intercept; $u_{i}$ and $v_{i}$ represent individual and time FEs, respectively; and $\mu_{it}$ is the spatial error term.

#### 3.2.3. Mediating Effect Model

The digital economy may affect haze pollution through technological innovation and the optimization and upgrading of the industrial structure. Therefore, we construct a mediating effect model and use a stepwise regression approach to verify this influence mechanism.
(4)$$lnM_{it}=\beta_{0}+\beta_{1}lnDE_{it}+\beta_{2}Controls_{it}+u_{i}+v_{t}+\mu_{it}$$
(5)$$lnPM_{2.5_{it}}=\gamma_{0}+\gamma_{1}lnDe_{it}+\gamma_{2}lnM_{it}+\gamma_{3}Controls_{it}+u_{i}+v_{t}+\mu_{it}$$
where, $M_{it}$ is mediating variable; other variables are the same as formula (1). Frist if $\alpha_{1}$, $\beta_{1}$ and $\gamma_{2}$ are significant, the mediating effect is significant. Second if $\alpha_{1}$, $\beta_{1}$ and $\gamma_{2}$ are significant but $\gamma_{1}$ insignificant which be named as complete meditating effect. Third, there is no mediating effect if $\alpha_{1}$ is insignificant.

#### 3.2.4. Spatial Durbin Model

Haze has liquid-like characteristics. Moreover, PM_2.5_ can remain in the atmosphere for a long time and be transported over a long distance. Thus, it is extremely difficult to eliminate. Therefore, we further explore the spatial effect of the digital economy on haze pollution. The SDM is a more general form of the spatial error model (SEM) and spatial lag model (SLM). The SDM includes two results: direct and indirect effects. We construct the SDM as follows:
(6)$$lnPM_{2.5_{it}}=\alpha_{0}+\rho WlnPM_{2.5_{it}}+\alpha_{1}lnDe_{it}+\varphi_{1}WlnDe_{it}+\alpha_{2}Controls_{it}+\varphi_{2}WControls_{it}+u_{i}+v_{t}+\mu_{it}$$
where ρ represents the coefficient of spatial autocorrelation; W is the spatial weight matrix; and $\varphi_{1}$ and $\varphi_{2}\;$ are the spatial interaction coefficients of the core interpretation variables and a series of control variables, respectively. The other variables are the same as in Formula (1).

### 3.3. Variable Description

#### 3.3.1. The Dependent Variable

The dependent variable of this paper is the PM_2.5_ concentration (PM_2.5_). PM_2.5_ is the main pollutant in haze pollution. We used raster data on the global average PM_2.5_ concentration monitored by satellites from the Atmospheric Composition Analysis Group Center of Dalhousie University, Canada (http://fizz.phys.dal.ca, accessed on 15 August 2022). Using ArcGIS 10.6 software, we parsed the annual average PM_2.5_ concentrations of prefecture-level cities in the middle and lower reaches of the Yellow River Basin from 2011 to 2019.

#### 3.3.2. The Core Independent Variable

The core independent variable in this study is the development level of the digital economy (DE). The digital economy takes digital knowledge and information as the key factors of production, digital technology as the core driving force, and modern information networks as an important carrier, through the deep integration of digital technology and the real economy, constantly improving the level of digitalization, networking, and intelligence of economic society, and accelerate the reconstruction of economic development and governance mode of a new economic form. Referring to Huang and Wang et al. [45] and considering the availability of data, we select five indicators from three systems to measure the level of digital economy development. The first system is digital infrastructure, including broadband Internet access users per 100 people and mobile phone users per 100 people. The second system is the digital industry (consisting of per capita total telecommunication services and the proportion of computer and software service personnel among the employees in urban units). The last system is digital finance, which is represented by the digital finance index. Firstly, the method of range standardization is used to non-dimensionalize the data. Secondly, the standardized data are further analyzed, and two principal components are determined according to the eigenvalues (>1) and the cumulative variance contribution rate. Again, according to the processing results, we calculate the principal component coefficient of the corresponding indicators; finally, the digital economy development level of each prefecture-level city is calculated according to the obtained coefficient. (Table 1).

#### 3.3.3. The Mediating Variables

Technological innovation (TI) and industrial structure (IS) are the mediating variables of this study. The number of invention patents in a region represents the advanced technology level of the region. Technological progress has a positive impact on haze pollution by changing the traditional production mode and eliminating outdated production capacity. The industrial structure is represented by the ratio of the added value of the tertiary industry to the added value of the secondary industry, which represents the level of premiumization and rationalization of the industry. The coefficient of the influence on haze pollution is negative.

#### 3.3.4. The Control Variables

Because many socioeconomic factors impact haze pollution, we selected the economic development level (PGDP), urban construction (UC), population density (PD), energy consumption (EC), urban greening (UG), and openness to the outside world (FDI) as control variables. GDP per capita is used to show the level of economic development. Urban construction refers to the ratio of the built-up urban area to the municipal area. RMB 10,000 standard coal is used as a measurement of energy consumption. Urban greening is usually indicated by the ratio of green coverage in the built-up urban area. Foreign direct investment is used to represent the level of openness to the outside world.

### 3.4. Data Sources

An empirical study was conducted using 2011–2019 panel data on 57 prefecture-level cities in the middle and lower reaches of the Yellow River Basin, as shown in Table 2. The data of the digital finance index were compiled by the Institute of Digital Finance, Peking University. The number of invention patents used to represent technological innovation comes from the China National Intellectual Property Administration. The other index data are available from the China City Statistical Yearbook. To eliminate the effect of heteroscedasticity caused by different data units, each data point was logarithmically processed.

Table 2 shows the descriptive statistical results of the reciprocal of variables, and calculates their skewness, kurtosis, and Jarque–Bera statistics. The results show that the average value of PM_2.5_ in 2011–2019 is 3.943, indicating that haze pollution is serious. The variance of the digital economy is 0.558, indicating that the regional differences in the development of the digital economy are large, and the proportion of regions with better development is large. For the mediating variable, the variance of technological innovation is 1.319, indicating that regional differences are large. For control variables, the regional differences in urban construction, energy consumption, and openness to the outside world are large, and other data differences are small. From the skewness, kurtosis, and Jarque–Bera statistics, except for greening, other data are stable.

## 4. Results

### 4.1. Spatiotemporal Evolution of the Digital Economy and PM_2.5_ Concentration

#### 4.1.1. Temporal Evolution

Using nonparametric KDE and MATLAB 2016 software, the kernel density curves of the digital economy level and PM_2.5_ concentration in the middle and lower reaches of the Yellow River Basin from 2011 to 2019 are drawn (Figure 2). From 2011 to 2019, the center of gravity of the digital economy shifted to the left and then to the right, indicating that the degree of the digital economy in the middle and lower reaches of the Yellow River Basin showed a fluctuating growth trend during the study period. The center of gravity of haze pollution slightly shifted to the right, demonstrating that the air quality in the middle and lower reaches of the Yellow River Basin improved from 2011 to 2019. The kernel density curves of the digital economy have only one peak, and the width of the peak increases, showing that there is no polarization in the level of the digital economy and that regional disparities are increasing. However, the tail on the right is longer than that on the left, indicating that the low-degree areas of the digital economy are clustered. The shape of the nuclear density curve of PM_2.5_ is wider but tends to be narrower, indicating that the regional difference of PM_2.5_ is large, but the difference is gradually narrowing. There were double peaks in 2014, 2015, 2016 and 2019, but the degree of polarization was weak. Since 2013, various provinces have successively implemented air pollution prevention, and the concentration of PM_2.5_ has gradually decreased as a whole. However, Shandong and Henan are dominated by traditional manufacturing industries. With the increase in production scale, air pollutants have gathered in the region, and the phenomenon of pollution polarization has begun to appear. With the gradual increase of governance, especially the priority development of communication technology in Shandong and Henan provinces, the transformation and upgrading of the manufacturing industry has been promoted, which has reduced resource consumption and pollutant emissions, and the polarization phenomenon has gradually weakened.

#### 4.1.2. Regional Spatial Evolution

The time scale of this study is 8 years. To better show the spatial evolution characteristics of the digital economy and the Yellow River Basin, and reflect the scientific of the research, the time nodes at both ends and in the middle (2011, 2015, and 2019) were selected. The spatial distribution and evolutionary characteristics of the digital economy and haze pollution in the middle and lower reaches of the Yellow River Basin were analyzed using ArcGIS 10.6 software. The development level of the digital economy is classified according to quartile classification. Based on China’s ‘ambient air quality standards’ (GB3095-2012) in the annual average concentration of PM_2.5_ limits, Zhou et al. [44] selected 35, 50, and 70 µg/m^3^ as the breakpoint to classify PM_2.5_ concentration.

Figure 3 shows the spatial evolutionary characteristics of the digital economy. Over the 9-year study period, the 57 cities showed an overall upward trend in the level of the digital economy, and the digital economy level of most cities was less than 0.1389 in 2011. However, in 2019, the cities with a digital economy level higher than 0.2589 accounted for 49%, and the number of cities with a digital economy level below 0.1389 was only two, Luohe and Tongchuan. The highest level of the digital economy by region was found in the lower reaches of the Yellow River Basin, especially Jinan, Qingdao, Yantai and Linyi in Shandong Province. In addition, some provincial capitals, including Zhengzhou, Taiyuan, and Xi’an, developed rapidly in terms of their digital economy level.

Figure 4 shows that from 2011 to 2019, the haze pollution situation improved in all 57 cities in the middle and lower reaches of the Yellow River Basin. Haze pollution was the most serious in Shandong and Henan. It was relatively serious in the northern cities of Shanxi and was relatively slight in most cities of Shaanxi. Overall, the PM_2.5_ concentration showed a spatial distribution pattern in which the PM_2.5_ concentration in the downstream region was more serious than that in the midstream region. In addition, the declining trend of PM_2.5_ concentrations varied by region. The declining trend in Shanxi was the largest, followed by Henan, especially Taiyuan and Zhengzhou, which experienced average annual declines of 6.281% and 5.966%, respectively. PM_2.5_ concentrations declined slightly in most cities in Shandong, especially Jining and Zaozhuang, where the average annual declines were only 2.620% and 2.336%, respectively.

### 4.2. Baseline Regression Estimation Results

Here, we used Stata15 software for regression analysis. The Hausman test rejected the null hypothesis, proving that the FE model is more suitable for this study. Since a two-way fixed effect model (Both) can exclude the endogeneity problems caused by uncontrollable time and individual factors, we chose the Both model to explore the impact of the digital economy on PM_2.5_ concentrations in the middle and lower reaches of the Yellow River Basin. Table 3 shows the baseline regression result; column (3) shows the regression results without the control variables, and column (4) shows the regression results adding a series of control variables. Regardless of whether the control variables are added, the digital economy has a significant (at the 5% or 1% level) negative impact on haze pollution in the middle and lower reaches of the Yellow River Basin, with elastic coefficients of 0.046 and 0.089, respectively. The regression results in column (4) show that for every 1% increase in the digital economy level, the PM_2.5_ concentration decreases by 0.89%.

In addition, we analyzed the effect of other control variables on PM_2.5_ concentration. The coefficient of lnPGDP is significantly negative (at the 1% confidence level), and the elasticity coefficient is −0.093. These results indicate that the level of economic development is beneficial for reducing haze pollution and that the PM_2.5_ concentration decreases by 9.3% for every 1% increase in per capita GDP. From the perspective of urban construction, there is a significantly positive correlation with the PM_2.5_ concentration, with a coefficient of elasticity of 0.027. With urban construction, the dust in the atmosphere increases, which is unfavorable for addressing haze pollution. The coefficient of population density is significantly positive, and the coefficient of elasticity is 0.204, which is harmful to reducing haze pollution. Energy consumption is significantly positive at the 1% confidence level (the elasticity coefficient is 0.050), which is not conducive to reducing the PM_2.5_ concentration. The effects of green coverage and foreign direct investment on the PM_2.5_ concentration are positive but not significant, indicating that they have little effect on addressing haze pollution.

### 4.3. Mechanism Verification

The above results show that the digital economy has a significant suppressive effect on haze pollution. However, we want to further explore the transmission mechanism of the effect of the digital economy on haze pollution. Table 4 considers the mediating effect of the digital economy on haze pollution through technological innovation (TI) and the industrial structure (IS). Column (1) indicates that the influence coefficient of the digital economy is 1.292 and significantly positively correlated with technological innovation (at the 1% confidence level), which means that the development of the digital economy is conducive to improving technological innovation. Column (2) shows that the regression coefficients of the digital economy and technological innovation on PM_2.5_ concentrations are significantly negative (−0.068 and −0.027, respectively). Column (3) shows that the digital economy is conducive to optimizing the industrial structure, and the regression coefficient is 0.154 (at the 5% confidence level). Column (4) indicates that the influence coefficient of the digital economy is negative and not significant, but the industrial structure has a significant negative correlation with the PM_2.5_ concentration, and the regression coefficient is −0.192 (at the 1% confidence level). In summary, technological innovation and industrial structure upgrading have a mediating effect. The digital economy not only directly suppresses haze pollution but also indirectly affects haze pollution through technological innovation and industrial structure upgrading.

### 4.4. Spatial Effect

#### 4.4.1. Spatial Correlation Test

Table 5 presents the spatial agglomeration of the digital economy and haze pollution in prefecture-level cities in the middle and lower reaches of the Yellow River Basin. From 2011 to 2019, Moran’s I of haze pollution was significantly positive between 0.703 and 0.835. Except in 2018 and 2019, Moran’s I of the digital economy was significantly positive between 0.005 and 0.038. In summary, the PM_2.5_ concentration and digital economy had spatial autocorrelation and spatial clustering phenomena.

Table 6 shows the test results of the spatial econometric model. The LM-lag, LM-Error and Robust LM-Error results were significant at the 1% level, indicating that spatial econometric analysis should be performed. The LR-lag and LR-error results reject the null hypothesis (at the 1% confidence level), indicating that the SDM cannot be degenerated into an SEM or SLM. Thus, we should select the SDM for regression analysis.

#### 4.4.2. Estimation of the Spatial Spillover Effect

Table 7 shows the SDM results based on the geographic adjacency matrix. To more accurately show the spatial spillover effect of the variables, the partial differential method is further adopted, and the relationship between the digital economy and haze pollution is divided into the direct effect, indirect effect and total effect. The direct effect represents the impact of the local digital economy on local haze pollution. The indirect effect reflects the impact of the digital economy in neighboring areas on local haze pollution. The total effect reflects the overall impact of the digital economy on haze pollution. Column (3) shows that the regression coefficient of the direct effect is −0.043 and is significantly negative (at the 1% confidence level). These results indicate that the digital economy can alleviate local haze pollution. In terms of the spatial spillover effect, the indirect effect is significantly negative with a coefficient of −0.278 and significant at the 1% confidence level, proving that the development of the digital economy in neighboring areas can effectively alleviate local haze pollution.

#### 4.4.3. Heterogeneity Analysis

In the research above, we found that the digital economy has a certain impact on haze pollution, but the local situation may be different from the overall spatial situation. Due to the different levels of economic development and the different industrial structures of cities in the middle and lower reaches of the Yellow River Basin, there are significant regional differences in the level of digital economy development and haze pollution. Based on the division standard of the Yellow River Water Conservancy Commission, Taohuayu in Zhengzhou city, Henan Province is the dividing point of the middle and lower reaches of the Yellow River Basin.

Table 8 shows the regression results of spatial heterogeneity in the middle and lower reaches of the Yellow River Basin. First, the direct, indirect and total effects in the middle and lower reaches of the basin are all significantly negative, which is consistent with the overall regression results. Second, in terms of the direct effect, the regression coefficients for the midstream and downstream regions are −0.048 and −0.063, respectively, and are significant at the 5% and 1% confidence levels, respectively. In terms of the indirect effect, the regression coefficient for the midstream region is −0.136 and is significant at the 10% confidence level. The regression coefficient for the downstream region is −0.217 and is significant at the 1% confidence level. Therefore, the contribution of the digital economy to haze pollution in the downstream region is much higher than that in the midstream region. The reason may be that the digital infrastructure in the downstream areas is more perfect, the digital technology is more developed, and the industry is more concentrated, which can just play the role of digital economy and significantly reduce haze pollution. The development level of the digital economy in the middle reaches is relatively low, with many high-emission enterprises and great pressure on ecological environment governance. Although the digital economy has played a certain role in suppressing haze pollution, the effect is relatively weak. Tian and Zhou et al.also reached a similar conclusion [46,47].

#### 4.4.4. Robustness Test

Table 9 shows the robustness test results. Two methods were adopted to demonstrate the robustness of the research results: (1) The first was replacing the spatial weight matrix. Specifically, we replaced the geographic adjacency matrix with a geographic distance matrix. The regression results were basically the same as the previous results, and both the direct effect and the indirect effect were significant at the 1% level (regression coefficients were −0.035 and −0.254, respectively). (2) The core explanatory variable was lagged by one period. We lagged the level of the digital economy by one period, and the results show that digital finance can significantly reduce haze pollution and have a spatial spillover effect. In conclusion, the results of this study are relatively robust.

From the perspective of some column control variables, urban construction has aggravated haze pollution, and the direct and indirect effects are significantly positive. The reason may be that the green space area is reduced during the rapid urban construction process, and a large amount of dust is generated, which deteriorates air quality. In addition, buildings are gathered closely, which is not conducive to the dissipation of pollutants. In addition, due to the heat island effect, suburban pollutants are brought back to the city, exacerbating pollution in urban areas, so the indirect effect is also significantly positive. Energy consumption is conducive to aggravating the haze pollution in the region, but it inhibits the aggravation of haze pollution in the adjacent areas. The reason is that the energy structure of the Yellow River Basin is dominated by coal, and energy consumption can produce a large number of pollutants. Pollutants spread to the adjacent areas, strengthen the appeal of residents’ pollution control, increase the intensity of governance, and alleviate the haze pollution in the adjacent areas. The direct effect of population density on haze pollution is significantly positive only in (2), and the indirect effect is positive but not significant. The economic activities and social activities of a high-density population will inevitably emit a large number of fine particles, aggravating haze pollution. The direct and indirect effects of foreign investment on haze pollution are only significantly positive in (2), which may follow the ‘pollution paradise’ hypothesis. In order to avoid the cost of environmental governance, developed countries transfer high-polluting industries to the region, exacerbating local haze pollution. The impact of other control variables is not significant.

## 5. Discussion

The digital economy is the main economic form after the agricultural economy and the industrial economy, and it holds great significance for environmental protection and sustainable development. Therefore, it is very important to explore the impact of the digital economy on haze pollution because doing so can provide a theoretical basis for the formulation of regional development goals. In the middle and lower reaches of the Yellow River Basin, the digital economy could effectively reduce haze pollution. This conclusion is similar to that drawn by Li et al. [48], who found that digital economy development was beneficial for reducing the PM_2.5_ concentration. Ecological environment protection and economic development are dialectical unity and indivisible. Therefore, haze governance and green development are also complementary. The impact of the digital economy on high-quality development reflects his green value. On the one hand, the digital economy of the Yellow River Basin has changed the input–output mode of traditional production through the development of a new generation of information technology, enhanced the ability of resource integration, and has a direct impact on ecological environment governance; on the other hand, the development of digital economy in the Yellow River Basin promotes the transformation of industrial structure dominated by labor-intensive industries and heavy industries to the rationalization stage and advanced stage, optimizes the allocation efficiency of social resources, and provides an important guarantee for pollution control [47,49]. For example, the digital transformation of Shandong Province has significantly promoted the total factor productivity of the manufacturing industry by improving innovation ability, optimizing human capital structure, and reducing costs [50]. It is also found that the development of digital technology in the Yellow River Basin has promoted the transformation of traditional manufacturing production technology and production methods, eliminated low-end pollution and backward production capacity through advanced technical standards, promoted clean production, and reduced pollutant emissions [51]. Therefore, technological innovation and industrial structure play an intermediary role in the relationship between the digital economy and haze pollution. In other words, the digital economy not only directly affected haze pollution but also indirectly alleviated haze pollution by promoting technological innovation and industrial structure upgrading. Zhou et al. [52] suggested that the digital economy is conducive to PM_2.5_ reduction through an advanced industrial structure. Wang et al. [53] found that digital finance can reduce haze pollution by promoting the mediating effect of innovation capability progress. These conclusions are similar to those of this study. The reasons are as follows: On the one hand, the digital economy reduces the proportion of traditional industries, optimizes the energy consumption structure, and improves the efficiency of resource allocation through the upgrading of the industrial structure to achieve cleaner production, the green transformation of enterprises, and a reduction in haze pollution. On the other hand, the development of a series of digital technologies, such as big data, blockchain and artificial intelligence, has laid the foundation for technological innovation. Technological innovation, especially green technological innovation, is an effective means of solving pollution problems. It can reduce haze pollution by promoting the clean production of enterprises and green consumption, reducing resource consumption and developing new energy. The mechanism of the influence of digital economy development on haze pollution is shown in Figure 5.

Meanwhile, some scholars believe that the impact of the digital economy on air quality is spatially heterogeneous [54]. Our research shows that the digital economy has a significant negative impact on haze pollution in the middle and lower reaches of the Yellow River Basin, but the impact effect in the lower reaches is greater than that in the middle reaches. In terms of the direct effect, the digital infrastructure in the downstream region, especially the Shandong section, is relatively sound, the development of the digital economy started earlier, and the integration of digital technology and traditional industries, digital industrialization and the industrial digital development guarantee system are relatively perfect, which can better release the dividends of the digital economy and play the role of digital empowerment. In terms of the indirect effect, the network level of the interior of the downstream region is higher, and digital access equipment covers a broader area, speeding up the sharing and transmission of information, knowledge and technology at the prefecture level. These factors are more likely to release the spatial spillover effect of digital economy development on haze pollution. In the middle reaches, the digital infrastructure and network level are relatively backward, and the transmission and sharing of data and information are relatively slow, which is not conducive to the spatial spillover effect of the digital economy.

## 6. Conclusions and Policy Implications

### 6.1. Conclusions 

Using 2011–2019 panel data on 57 prefecture-level cities in the middle and lower reaches of the Yellow River Basin, this study evaluates the development level of the digital economy and PM_2.5_ concentration. Then, it adopts an FE model, a mediating effect model and the SDM to empirically study the impact mechanism and spatial spillover effect of the digital economy on haze pollution. The main four conclusions are as follows.

First, the development level of the digital economy showed an overall upward trend, and high-degree areas were mainly distributed in cities such as Jinan and Qingdao in Shandong Province or other provincial capitals. The overall concentration of PM_2.5_ showed an overall downward trend, and highly polluted areas were mainly distributed in the cities of Shandong and Henan Provinces.

Second, the impact of the digital economy on haze pollution is negative. With the development of the digital economy, air quality improves, and the robustness test confirms this conclusion.

Third, the digital economy can not only directly affect haze pollution but can also affect haze pollution through technological innovation and optimization of the industrial structure.

Finally, the digital economy has a spatial spillover effect on haze pollution. The digital economy not only inhibits haze pollution in a region but also facilitates mitigation in neighboring areas. Furthermore, the impact of the digital economy on haze pollution has regional heterogeneity, and the effect in the downstream region is significantly higher than that in the midstream region.

This paper discusses the influence mechanism and spatial spillover effect of the digital economy on haze pollution. However, there are still some limitations in this paper. For example, due to data limitations, this paper selects only five indicators from the digital infrastructure, digital industry and digital finance dimensions to measure the level of the digital economy. Future research will focus on collecting data from multiple measurement indicators to further improve the digital economy index system.

### 6.2. Policy Implications

Based on the conclusions above, this paper focuses on further improving the action mechanism of the digital economy and on alleviating haze pollution, the following policy implications are proposed.

First, a high level of digital economy development is the key to tackling haze pollution. Specifically, the government should strengthen the construction of new infrastructure, such as 5G base stations, and realize continuous coverage of areas above the township level and important roads. We will improve the environment for the development of the digital economy, carry out comprehensive publicity on digital development, encourage governments at all levels and industries and sectors to firmly adopt the concept of digital development, and create a favorable atmosphere for all of society to participate in and support digital development.

Second, the innovative development of digital technology promotes clean production and helps alleviate haze pollution. On the one hand, the government should strengthen talent training in the digital economy, promote cooperation between critical enterprises of the digital economy and scientific research institutes, cultivate compound and practical digital talent, and provide a talent guarantee for developing the digital economy. On the other hand, we can guide capital, resources, skill, and industries to gather in digital fields, actively cultivate regional innovation efficiency and innovation capacity, develop clean production technologies, and promote pollution and emission reduction.

Third, industrial digital transformation is an important mechanism to improve energy and resource utilization efficiency and reduce PM_2.5_ emissions. Governments should promote the digital transformation of agriculture, manufacturing and services. The government should promote the application of big data, internet and artificial intelligence in agricultural production, operation and management; promote the restructuring of the traditional manufacturing system, change the driving force, and realize the transformation of high-end, green services; promote the integration and penetration of digital technologies with producer and consumer services; and foster new service industries such as new retail, the sharing economy and the platform economy.

Finally, according to the regional differences in haze pollution and the stage characteristics of digital economy development, there should be formulation of digital economy development strategies that are conducive to reducing PM_2.5_ concentration according to local conditions. Regions with a relatively weak digital economy level should strengthen the construction of digital infrastructure, and regions with high level of digital economy should continue to deepen the development of the digital economy to play an innovation-leading and radiation-driving role to narrow regional differences.

## Figures and Tables

**Figure 1 ijerph-19-17094-f001:**
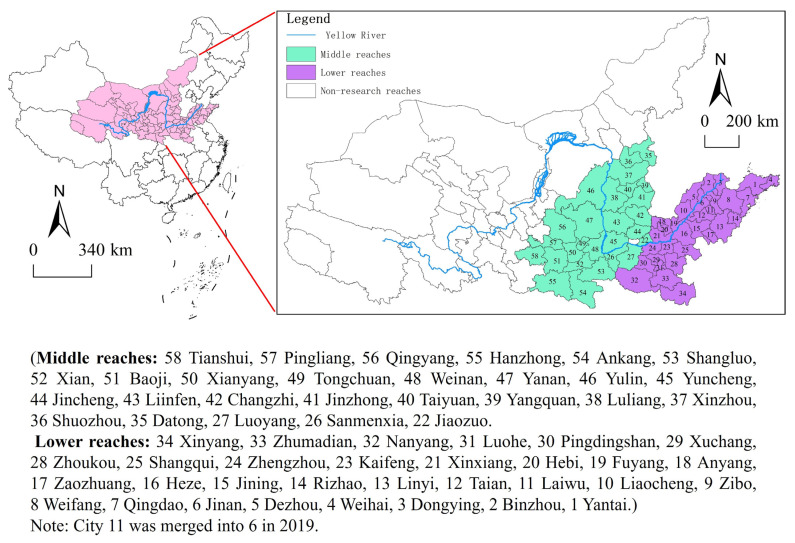
Study area.

**Figure 2 ijerph-19-17094-f002:**
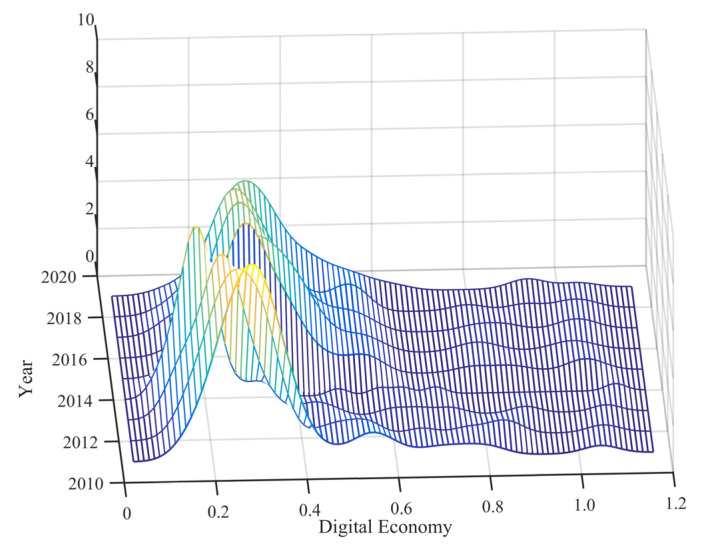
Kernel density estimation of the digital economy level and PM_2.5_ concentration.

**Figure 3 ijerph-19-17094-f003:**
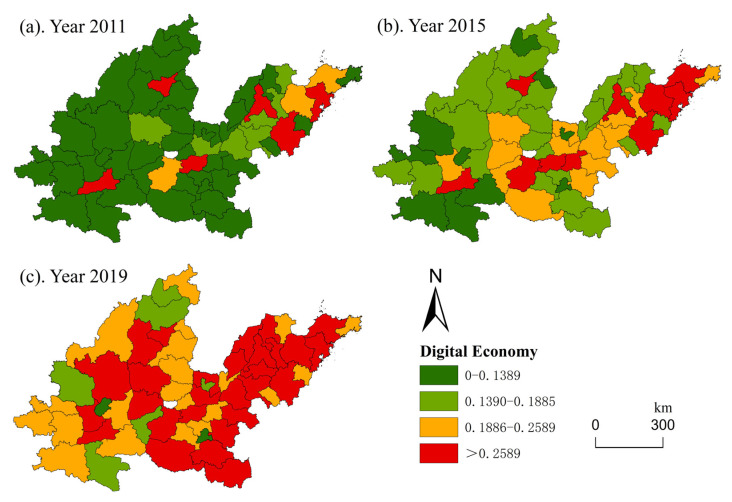
Spatial distribution of the digital economy level.

**Figure 4 ijerph-19-17094-f004:**
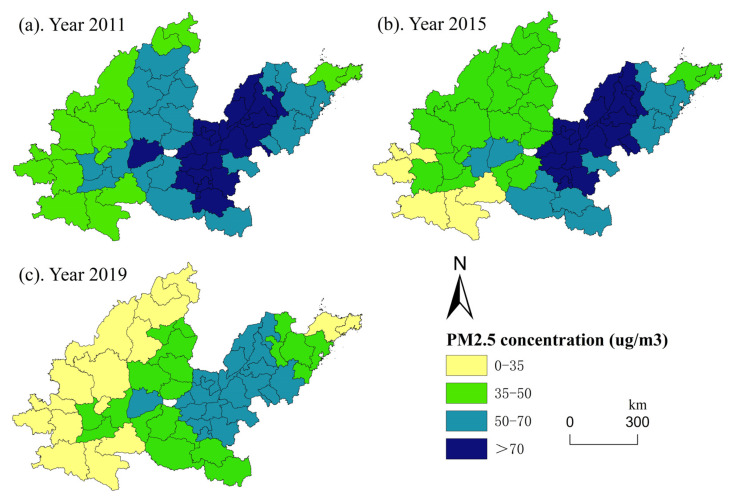
Spatial distribution of PM_2.5_ concentrations.

**Figure 5 ijerph-19-17094-f005:**
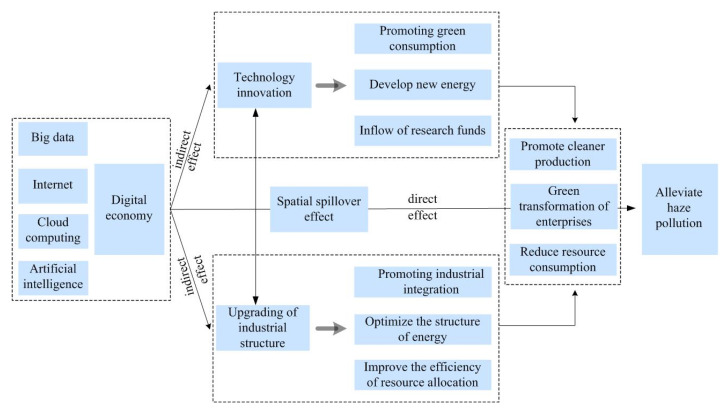
The mechanism of the effect of the digital economy on haze pollution.

**Table 1 ijerph-19-17094-t001:** Measurement system of the digital economy.

	System Level	Indicator Level	Score Coefficient	Weight of Index
Digital economy development	Digital infrastructure	Internet broadband access users per 100 people	0.354	0.231
		Mobile phone users per 100 people	0.381	0.248
	Digital industry	Per capita total telecommunication services	0.069	0.045
		The proportion of computer and software service personnel among the employees in urban units	0.392	0.255
	Digital finance	Digital finance index	0.339	0.221

**Table 2 ijerph-19-17094-t002:** Descriptive statistics of the variables.

Type of Variable	Variable	Obs	Mean	SD	Min	Max	Skewness	Kurtosis	JB
Dependent Variable	lnPM_2.5_	513	3.943	0.306	3.203	4.523	−0.228	2.165	19.352
Independent Variable	lnDE	513	−1.649	0.558	−3.643	0.151	0.200	3.672	13.063
Intermediate Variables	lnTI	513	7.181	1.319	3.638	10.568	0.129	2.728	3.010
	lnIS	513	−0.244	0.445	−1.364	1.236	−0.070	2.964	0.444
Control Variables	lnPGDP	513	10.661	0.511	9.299	12.165	0.193	2.900	3.395
	lnUC	513	1.857	1.068	−0.799	4.541	−0.205	2.548	7.960
	lnPD	513	6.022	0.738	4.142	7.450	−0.573	2.339	37.365
	lnEC	513	4.734	1.114	0.822	7.368	−0.480	3.155	20.239
	lnUG	513	3.640	0.369	−0.528	4.104	−8.541	87.353	158,329.127
	lnFDI	513	9.877	1.717	4.635	13.708	−0.355	2.791	11.704

**Table 3 ijerph-19-17094-t003:** Results of the baseline regression estimation.

Variables	(1) RE	(2) FE	(3) Both	(4) Both
lnDE	−0.172 *** (−8.17)	−0.171 *** (−8.53)	−0.046 ** (−2.53)	−0.089 *** (−3.80)
lnPGDP	−0.225 *** (−7.48)	−0.253 *** (−8.37)		−0.093 *** (−3.09)
lnUC	0.0416 *** (3.36)	0.001 (0.12)		0.027 *** (3.37)
lnPD	0.144 *** (5.79)	−0.052 * (−1.75)		0.204 *** (10.87)
LnEC	−0.042 *** (−4.96)	−0.046 *** (−6.25)		0.050 *** (3.06)
lnUG	0.023 (1.61)	0.019 (1.55)		0.018 (0.84)
lnFDI	0.013 *** (3.40)	0.005 (1.35)		0.007 (0.95)
Constant	5.090 *** (14.14)	6.767 *** (18.80)	4.007 *** (99.16)	3.246 *** (11.94)
R^2^	0.233	0.016	0.132	0.712
N	513	513	513	513

Note: ***, **, and * indicate significance at the 1, 5, and 10% levels, respectively.

**Table 4 ijerph-19-17094-t004:** Results of the mechanism verification analysis.

Variables	(1) lnTI	(2) lnPM_2.5_	(3) lnIS	(4) lnPM_2.5_
lnDE	1.292 *** (19.55)	−0.068 *** (−3.85)	0.154 ** (2.10)	−0.012 (−0.56)
lnTI		−0.027 *** (−3.34)		
lnIS				−0.192 *** (−8.31)
Constant	0.203 (0.25)	4.214 *** (17.02)	5.478 *** (3.05)	3.806 *** (14.93)
Control	Yes	Yes	Yes	Yes
Year fixed effect	Yes	Yes	Yes	Yes
City fixed effect	Yes	Yes	Yes	Yes
R^2^	0.850	0.023	0.275	0.734
*N*	513	513	513	513

Note: ***, and ** indicate significance at the 1, and 5% levels, respectively.

**Table 5 ijerph-19-17094-t005:** Moran’s I of the digital economy and the PM_2.5_ concentration.

Year	PM_2.5_	DE	Year	PM_2.5_	DE
2011	0.723 ***	0.027 *	2016	0.749 ***	0.021 *
2012	0.739 ***	0.038 **	2017	0.703 ***	0.017 *
2013	0.798 ***	0.016 *	2018	0.735 ***	0.005
2014	0.760 ***	0.038 **	2019	0.747 ***	0.008
2015	0.835 ***	0.018 *			

Note: ***, **, and * indicate significance at the 1, 5, and 10% levels, respectively.

**Table 6 ijerph-19-17094-t006:** Test results of the spatial econometric model.

Spatial Autocorrelation Text	Statistic
LM-lag	7.028 ***
Robust LM-lag	0.039
LM-Error	99.170 ***
Robust LM-Error	92.180 ***
LR-lag	23.560 ***
LR-Error	109.590 ***

Note: *** indicate significance at the 1% levels, respectively.

**Table 7 ijerph-19-17094-t007:** The results of the spatial spillover effect.

Variables	(1) SEM	(2) SLM	(3) SDM
lnDE	−0.0168 *** (−2.00)	−0.031 *** (−4.62)	−0.021 ** (−2.49)
W × lnDE			−0.018 (−1.58)
Direct			−0.043 *** (−3.80)
Indirect			−0.278 *** (−3.72)
Total			−0.321 *** (−3.90)
Control	Yes	Yes	Yes
Time fixed effect	Yes	Yes	Yes
City fixed effect	Yes	Yes	Yes
R^2^	0.368	0.278	0.449
Log-likelihood	822.429	865.443	877.224
*N*	513	513	513

Note: ***, and ** indicate significance at the 1, and 5% levels, respectively.

**Table 8 ijerph-19-17094-t008:** The results of heterogeneity analysis.

Variables	(1) Midstream Region	(2) Lower Region
Direct	−0.048 ** (−2.48)	−0.063 *** (−3.61)
Indirect	−0.136 * (−1.69)	−0.217 *** (−2.44)
Total	−0.184 * (−1.95)	−0.279 *** (−2.74)
Control	Yes	Yes
Time fixed effect	Yes	Yes
City fixed effect	Yes	Yes
R^2^	0.186	0.019
Log-likelihood	502.024	589.282
N	243	270

Note: ***, **, and * indicate significance at the 1, 5, and 10% levels, respectively.

**Table 9 ijerph-19-17094-t009:** The results of the robustness test.

Variables	(1) Changing the Spatial Weight Matrix			(2) Lagging DE One Period		
	Main Effect	Direct Effect	Indirect Effect	Main Effect	Direct Effect	Indirect Effect
lnDE	−0.012 (−1.27)	−0.035 *** (−2.17)	−0.254 *** (−3.33)			
W × lnDE	−0.030 ** (−2.22)					
L.lnDE				−0.032 *** (−3.37)	−0.048 *** (−3.62)	−0.215 ** (−2.14)
W × L.lnDE				0.001 (0.04)		
lnPGDP	−0.007 (−0.39)	−0.013 (−0.69)	−0.069 (−0.59)	−0.008 (−0.49)	−0.024 (−1.16)	−0.186 (−1.21)
lnUC	0.010 ** (2.13)	0.031 *** (3.45)	0.231 *** (2.92)	0.011 *** (2.56)	0.021 ** (2.23)	0.117 (1.25)
lnPD	0.005 (0.76)	0.009 (0.47)	0.039 (0.32)	0.018 * (1.72)	0.031 * (1.72)	0.174 (1.13)
LnEC	0.008 ** (2.43)	0.002 (0.37)	−0.075 *** (−2.95)	0.006 * (1.78)	0.002 (0.41)	−0.055 * (−1.80)
lnUG	0.004 (0.76)	0.007 (0.89)	0.025 (0.62)	0.001 (0.18)	0.014 (1.48)	0.156 * (1.84)
lnFDI	0.000 (0.23)	0.003 (1.10)	0.026 (1.61)	0.391 (1.51)	0.004 * (1.68)	0.045 ** (2.15)
Time fixed effect	Yes			Yes		
City fixed effect	Yes			Yes		
R^2^	0.584			0.608		
Log-likelihood	851.836			775.885		
N	513			456		

Note: ***, **, and * indicate significance at the 1, 5, and 10% levels, respectively.

## Data Availability

The data that support the findings of this study are available upon request from the corresponding author.

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
