# Peer review of "Impact of the Digital Economy on PM2.5: Experience from the Middle and Lower Reaches of the Yellow River Basin"

_ijerph, 2022, doi:10.3390/ijerph192417094_

Round 1

Reviewer 1 Report

The authors used raster data and panel data to explored the impact of the digital economy on PM2.5, which provide great significance for alleviation haze pollution. However, there still have some problems need to be revised.

Major problems,

1.Clarify the research object. The title is explored the influence of the digital economy on PM2.5, while the authors illustrated the paper focuses on the multidimensional path of the effect of the digital economy on carbon emissions in line 66-68. The PM2.5 is different from carbon emission.

2.The presentation is lack of logic. For instance, the authors illustrated the digital economy level and PM2.5 concentration in the downstream region are higher than those in the middle region, and the digital economy is negatively correlated with haze pollution. The high digital economy level should correspond to low PM2.5, while the digital economy level of downstream region is higher than middle region, the PM2.5 concentration is also higher than middle region.

3.How to define digital economy in this manuscript?

4.In literature review, the authors listed the content references simply. It is lack of summarizing. In addition, the authors’ last name of references should not be shown in text.

5.In line 239-241, the authors should introduce the method of calculating digital economy development detailed. For example, the weight of each indicator.

6.The PM2.5 should be mentioned in each of policy implication.

Minor problems,

1.The authors should cite the original method paper rather than applied method paper in model setting section.

2.The title of 3.3 is the same as 3.4.

3.The website of PM2.5 concentration data source should be supplied.

4.In line 265, the authors mentioned the number of invention patents. However, it doesn’t be shown in index system.

5.The version of ArcGIS and method of classification should be illustrated.

6.Which software are the authors use to calculate the baseline regression?

7.The conclusions should be divided into several points.

8.The limitation(490-495) should be moved to section 6.

Reviewer 2 Report

The manuscript entitled “Impact of the Digital Economy on PM2.5: Experience from the Middle and Lower Reaches of the Yellow River Basin” (ijerph-2079744) was aimed at measuring the impact of the ‘digital economy’ (as defined by the authors) on haze pollution in the Middle and Lower Reaches of the Yellow River Basin, by resorting to various econometric techniques.

To improve the quality of ijerph-2079744, I propose the following:

(1) Further develop Table 2 with Skewness, Kurtosis and Jarque-Bera. Finish Section 3 with a brief discussion regarding the results from the descriptive statistics of the variables. 

(2) Regarding the temporal evolution presented in subsection 4.1.1 – I suggest discussing the possible reasons why the kernel density curves of PM2.5 have two peaks in 2014, 2015, 2016 and 2019; as well as tapping into a commentary on risk mitigation strategies in the relation of digital economy–pollution.

(3) More explanation is needed regarding the motives behind selecting the years 2011, 2015, 2019 for the analysis in subsection 4.1.2.

(4) More reflexivity is required regarding the other factors that could have positively impacted the pollution outcome, other than the evolution of the digital economy. This could be discussed in the robustness subsection (4.4.4) and it would add more value to the already existing econometric tests carried out in this part of the paper.

(5) This paper needs more practical examples regarding the contribution of the digital economy on improving pollution outcomes. More specially, the authors should mention what specific aspects (examples) of the digital economy triggered a reduction of PM2.5 pollution in the Middle and Lower Reaches of the Yellow River Basin.

Reviewer 3 Report

Thank you for giving me this opportunity to read the manuscript entitled "Impact of the Digital Economy on PM2.5: Experience from the Middle and Lower Reaches of the Yellow River Basin". The topic of this manuscript is interesting and would be a good contribution to this field. I think it could be considered for publication in IJERPH once the following issues are addressed.

1.     Please replace the keywords that already appear in the manuscript's title with close synonyms or other keywords, which will also facilitate your paper being searched by potential readers.

2.     "Scale", and "Compass" should be added to the sub-map in Figure 1. Besides, please enlarge the text of Figure 1 to make sure it can be read clearly.

3.     Line 85, “… health effects [9]…”: a paper titled “Dynamic assessment of PM2. 5 exposure and health risk using remote sensing and geo-spatial big data” is suggested to be added as a reference.

4.     Data sources of the control variables introduced in Section 3.3.4 should be provided.

5.      Some grammatical errors exist in the manuscript. Therefore, a critical review of the manuscript's language will improve its readability.

Round 2

Reviewer 2 Report

The authors have improved the manuscript according to the reviewer comments.

There are some typing errors in the new text written in red.

Author Response

Response to Reviewer 2 Comments

Thank you so much for reading our article carefully and giving positive comments. According to your constructive suggestions, we have made extensive corrections to our previous draft and added the necessary material and analysis to supplement our results. We expect that the revised manuscript could meet your requirements. In case any additional revisions are needed, we are ready to embark on these. The detailed modifications are listed below.

Point 1: There are some typing errors in the new text written in red.

Response 1: Thanks for your constructive suggestion. According to your comments, we have carefully examined and modified the red font part of the manuscript, mainly modifying the spelling errors of words, the application errors of prepositions, etc. Please check the red font part of the manuscript.

For example :

The digital economy takes digital knowledge and information as the key factors of production, digital technology as the core driving force, and modern information networks as an important carrier, through the deep integration of digital technology and the real economy, constantly improving the level of digitalization, networking, and intelligence of economic society, and accelerate the reconstruction of economic development and governance mode of a new economic form.